# Peer review of "SM-GCG: Spatial Momentum Greedy Coordinate Gradient for Robust Jailbreak Attacks on Large Language Models"

_electronics, doi:10.3390/electronics14193967_

Round 1

Reviewer 1 Report

Comments and Suggestions for Authors

Manuscript Title: SM-GCG: Spatial Momentum Greedy Coordinate Gradient for Robust Jailbreak
Attacks on Large Language Models

Comment: The authors propose a novel jailbreak attack method, Spatial Momentum Greedy Coordinate
Gradient (SM-GCG), which introduces a spatial momentum mechanism to enhance the gradient
estimation process in adversarial prompt optimization. The topic is timely. However, to further
strengthen the manuscript, the following points should be addressed.

1. This paper introduces the "Spatial Momentum". However, in both the abstract and
contributions sections, while claiming "a relative improvement of up to 50%," it fails to
explicitly specify the computational basis or comparative benchmarks underlying this
quantitative assertion.
2. The article provides an abstract description of specific transformation operations for "multi-
space gradient sampling," such as synonym substitution in text space and token displacement
operations in token space, lacking concrete examples or algorithmic pseudocode.
3. The article mentions that only 100 malicious prompts were used for the experiment due to
limitations in computing resources, but it does not clarify whether these 100 prompts are
representative or how their diversity is ensured. It is necessary to provide a detailed description
of the specific method for selecting the subset and discuss the potential impact of sample bias.
4. The article points out that SM-GCG still has limitations in terms of metastasis, but it fails to
deeply analyze the reasons or propose improvement directions.
5. Figure 3 shows the comparison of the loss curves between GCG and SM-GCG. However, the
article merely briefly mentions "reducing oscillations and converging to a lower loss", without
explaining the specific mechanism of the reduction in oscillations or its direct correlation with
the attack success rate. Add explanations of the key features of the curves in the figure caption
or the main text.
6. The text alternates the use of terms such as "spatial momentum" and "multi-space gradient
sampling", which may cause confusion. It is recommended to clearly define the core terms for
the first time they are used and maintain consistency throughout the text.

Author Response

We thank the reviewers for their insightful comments and constructive suggestions, which have helped us to significantly improve the manuscript.

Comments 1: This paper introduces the "Spatial Momentum". However, in both the abstract and contributions sections, while claiming "a relative improvement of up to 50%," it fails to explicitly specify the computational basis or comparative benchmarks underlying this quantitative assertion.

Response 1: Agree. We have, accordingly, revised the claims regarding "a relative improvement of up to 50%" in both the abstract and contributions sections. Specifically, we have modified the statement to indicate an improvement in attack success rate of 10%-15% over the baseline method GCG. These changes can be found on page 3, lines 87 in the revised manuscript.

Comments 2: The article provides an abstract description of specific transformation operations for "multi-space gradient sampling," such as synonym substitution in text space and token displacement operations in token space, lacking concrete examples or algorithmic pseudocode.

Response 2: Agree. We have, accordingly, added concrete implementation examples following the abstract descriptions for each space to enhance clarity and practical understanding. The changes can be found on Page 9, Lines 296-302, Page 10, Lines 319-341, Page 11, Lines 356-365, Page 11, Lines 383-386 and Page 12, Lines 391.

Comments 3: The article mentions that only 100 malicious prompts were used for the experiment due to limitations in computing resources, but it does not clarify whether these 100 prompts are representative or how their diversity is ensured. It is necessary to provide a detailed description of the specific method for selecting the subset and discuss the potential impact of sample bias.

Response 3: Agree. We have, accordingly, revised the "Datasets" section to address this concern. Specifically, we have added a detailed description of our stratified random sampling method used to select the representative subset of 100 malicious prompts from the AdvBench dataset. We explain that we used the original category labels as strata and selected prompts proportionally to each category's size in the full dataset. Additionally, we now include a discussion of the potential impact of sample bias, acknowledging two main limitations: the possible underrepresentation of rare categories and the incomplete capture of intra-category linguistic diversity. We also note that despite these limitations, our sampling approach provides a robust and practical compromise. These changes can be found in the "Datasets" section on Page 13, Lines 452-471.

Comments 4: The article points out that SM-GCG still has limitations in terms of metastasis, but it fails to deeply analyze the reasons or propose improvement directions.

Response 4:  Agree. We have, accordingly, revised the Discussion section to thoroughly analyze the reasons behind these limitations and propose concrete directions for future improvements. Specifically, we have added an in-depth explanation identifying two primary reasons for the transferability limitations: (1) the fundamental architectural and alignment differences between source and target models that create distinct decision boundaries, and (2) the inherent bias in the optimization process toward the local geometry of the source model, despite mitigation by spatial momentum. Furthermore, we explicitly propose two promising research directions for future work: (a) incorporating model-agnostic constraints or ensemble-based optimization during attack generation, and (b) leveraging insights from spatial momentum vectors to identify model-invariant vulnerable features. These revisions can be found in the revised manuscript on Page 16, line 580-593.

Comments 5: Figure 3 shows the comparison of the loss curves between GCG and SM-GCG. However, the article merely briefly mentions "reducing oscillations and converging to a lower loss", without explaining the specific mechanism of the reduction in oscillations or its direct correlation with the attack success rate. Add explanations of the key features of the curves in the figure caption or the main text.

Response 5: Agree. We have, accordingly, revised the manuscript to comprehensively address the key features of the loss curves. Specifically, we have added explanations for: 1) the dynamical mechanism behind the amplitude reduction of oscillations; 2) the association between the smooth convergence phase and the improved generalization capability of the resulting adversarial suffixes; 3) the quantitative impact of the lower loss plateau on the attack success rate, supported by the calculation of the Spearman's rank correlation coefficient between the loss and the attack success rate across ten independent runs with varying configurations, along with the inclusion of a representative line plot for intuitive visualization. Furthermore, the original lengthy figure caption has been streamlined, with its detailed experimental setup moved into the main text for better readability. These changes can be found on Page 6, Lines 220-244.

Comment 6: The text alternates the use of terms such as "spatial momentum" and "multi-space gradient sampling", which may cause confusion. It is recommended to clearly define the core terms for the first time they are used and maintain consistency throughout the text.

Response 6: Thank you for your valuable feedback. To clarify, "spatial momentum" refers to the mathematical and dynamic interpretation, while "multi-space gradient sampling" denotes the method practically applied in the algorithm. We intend to retain both terms but will ensure their definitions are explicitly stated upon first use and consistently maintained throughout the text.

Reviewer 2 Report

Comments and Suggestions for Authors

This paper presents an enhanced Greedy Coordinate Gradient method, named SM-GCG, for robust jailbreak attacks on large language models. By using spatial momentum that aggregates gradient information from multiple semantically equivalent transformation spaces, SM-GCG can accurately estimate the global gradient direction and therefore achieve a higher jailbreak attack success rate. The authors evaluated the proposed method and compared its performance regarding ASR and Recheck with several benchmarks. Experimental results show that SM-GCG generally outperforms the other methods in the white-box attack settings. Ablation study and transferability study were also conducted to highlight the interpretability and the limitation of the proposed method.

Overall, the paper is structured. The research fits the scope of the Electronics journal. The references are appropriate and up to date. The topic should be interesting to certain readers.

Proofreading is required before re-submission. The authors may consider using software (e.g., Grammarly, Writefull) to assist polishing the manuscript. 

Author Response

We thank the reviewers for their insightful comments and constructive suggestions, which have helped us to significantly improve the manuscript.

Comments 1: Proofreading is required before re-submission. The authors may consider using software (e.g., Grammarly, Writefull) to assist polishing the manuscript.
Response 1: Agree. We have, accordingly, revised the manuscript thoroughly to improve the clarity and quality of the English language. The text has been carefully proofread and polished using recommended tools to ensure grammatical accuracy and better expression. These language refinements can be found throughout the revised manuscript.

Reviewer 3 Report

Comments and Suggestions for Authors
  1. There are 21 references (almost 50% of the listed references) are non-peer reviewed (references [5], [7], [8], [9]. [12], [13], [14], [15], [16], [18], [22], [29], [30], [33], [34], [35], [36], [37], [39], [40], [41]), actually, the arguments of the non-peer reviewed references cannot be adopted as a trusted source of information. Hence, please replace them with peer-reviewed alternatives. Additionally, reference [5] seems to be not directly related to the study.
  2. The importance of the provided work and the analytical connection to the related studies should be emphasized in the introduction.
  3. The outlines for the rest of the article should be provided in the last paragraph in the introduction section.
  4. The “Previous Research” section should be more analytical instead of listing what is there, it lacks the critical analysis of the listed studies.
  5. The methodology is detailed; however, it should be augmented with full complexity analysis of the provided model. Additionally, it should provide some examples of prompt transformation before and after. It should indicate the rationale behind the settings of the different assumptions in the provided model and the hyper-parameters used later in the Results section and how the hyperparameters are tuned. Also, the white-box assumption seems to be unrealistic for closed-source models. Additionally, the Transferability should be further investigated since cross-model transfer is not fully analyzed.
  6. The results section should provide some experiments for larger samples than that already presented. Additionally, other models should be compared since the models used for the comparison are only open-source, it will be important to test the performance of the provided models against some closed-source models such as GPT 4. Moreover, there are very recent models that have been highlighted in the literature such as JailPO that will be worth to be compared against the performance of the provided model. Also, many parameters are missing such as the stopping criteria, the details of the hyperparameters … etc., which mainly limit the reproducibility of the results and the cross-model transfer is not fully tested. The complexity of the provided model should be compared against those for the benchmarks used.
  7. The results section should be more balanced between being descriptive and being analytical, it is mainly descriptive in current version of the manuscript.
  8. The limitations of the presented approach should be discussed in detail in the discussion section with some insights into future directions.
  9. The caption of Fig. 3 is really long. Additionally, the titles of Tables 2 and 4 are really vague.
  10. Section 3: “Materials and Methods” better name it as “Methodology”.
Comments on the Quality of English Language
  1. Line 97: “prompt construction[14–18],” should be “prompt construction [14–18],” please handle similar occurrence in the paper.
  2. Better not to use first person tone (e.g., we, our … etc.) in academic writing and use passive action instead.
  3. Line 5: “promise,their” should be “promise, their”.
  4. Line 33: “prompts with malicious question” should be “prompts with malicious questions” or “prompts with a malicious question”.
  5. Line 43: “Fig 1.” Should be “ 1.” Also, please correct similar occurrences for the listing of the other figures.
  6. There are many sentences in the article that are too long to be understood, for example, the sentence starting in line 256, which limits the readability. Please consider separating such sentences into multiple ones.
  7. Line 273: “where W is embedding weight matrix” should be “where W is the embedding weight matrix”.
  8. Line 344: “typically include phrases like” the word “like” when means similar to is a bit unformal for academic writing, please kindly use “similar to” or “such as” … etc. that will be more formal.
  9. Line 172: “GCG’s gradient” should be “GCG gradient”, the “ ’s” is mainly used with humans.
  10. The paper needs full proofreading.

Author Response

We thank the reviewers for their insightful comments and constructive suggestions, which have helped us to significantly improve the manuscript.

Comments 1: There are 21 references (almost 50% of the listed references) are non-peer reviewed (references [5], [7], [8], [9]. [12], [13], [14], [15], [16], [18], [22], [29], [30], [33], [34], [35], [36], [37], [39], [40], [41]), actually, the arguments of the non-peer reviewed references cannot be adopted as a trusted source of information. Hence, please replace them with peer-reviewed alternatives. Additionally, reference [5] seems to be not directly related to the study.

Response 1: Thank you for this valuable feedback. We acknowledge the concern regarding the use of non-peer-reviewed references. Since our research focuses on a relatively new field, we aimed to incorporate the most recent and relevant findings to support our work. In response to your comments, we have replaced a portion of the non-peer-reviewed references with appropriate peer-reviewed alternatives. Reference [5], which was originally used to demonstrate the capabilities of large language models in code generation, has been removed as it was not directly related to the core study.

Comments 2: The importance of the provided work and the analytical connection to the related studies should be emphasized in the introduction.

Response 2: Agree. We have, accordingly, revised the introduction to emphasize the importance of our work and its analytical connection to related studies. Specifically, we have added a new paragraph that provides a detailed analysis of the mechanisms, contributions, and limitations of existing works such as MAC. This paragraph explicitly highlights their connection to our study (i.e., attempting to improve GCG) and key distinctions (e.g., our use of spatial momentum). Additionally, we have further underscored the significance of our work by clarifying the need for a fundamental shift in optimization strategy. Finally, we have included an outline of the remaining sections to enhance the overall structure. These changes can be found in the revised manuscript on Page 2, Lines 52-59.

Comments 3: The outlines for the rest of the article should be provided in the last paragraph in the introduction section.

Response 3: Agree. We have, accordingly, revised the introduction section to include an outline of the paper’s structure. The revised text can be found on page 4, lines 103-110.

Comments 4: The “Previous Research” section should be more analytical instead of listing what is there, it lacks the critical analysis of the listed studies.

Response 4: Agree. We have, accordingly, revised the "Previous Research" section to incorporate critical analysis for each category of methods discussed. Specifically, after introducing white-box attacks, we have added a critique highlighting their primary limitations, such as strong assumptions about model accessibility and the tendency of some methods to produce semantically incoherent suffixes vulnerable to perplexity-based detection. Similarly, following the description of black-box attacks, we now discuss their practical constraints, including dependence on specific linguistic loopholes, high query overhead, and computational costs. For defense strategies, we have expanded the analysis to point out the reliance on heuristics that can be circumvented and the trade-offs between safety and general capabilities. Finally, we have added a concluding paragraph that synthesizes the overall state of the field, emphasizing the academic nature of current approaches and the gap in practicality for real-world deployment. These changes can be found on page 4, lines 127 - page 5, lines 185.

Comments 5: The methodology is detailed; however, it should be augmented with full complexity analysis of the provided model. Additionally, it should provide some examples of prompt transformation before and after. It should indicate the rationale behind the settings of the different assumptions in the provided model and the hyper-parameters used later in the Results section and how the hyperparameters are tuned. Also, the white-box assumption seems to be unrealistic for closed-source models. Additionally, the Transferability should be further investigated since cross-model transfer is not fully analyzed.

Response 5: Agree. We have, accordingly, revised the manuscript to address these points. First, we have augmented Appendix D with a full complexity analysis of the models used in the study. This analysis quantitatively evaluates the computational complexity and parameter scale, highlighting the high consistency among the models due to their shared LLaMA-7B foundation architecture. This addition can be found on Page 19, Lines 691-704.

Second, to clarify the prompt transformation process, we have added concrete examples illustrating the transformations applied in each of the five perturbation spaces (Candidate, Text, Token, One-hot, Embedding). These descriptions are located in Page 9, Lines 296-302, Page 10, Lines 319-341, Page 11, Lines 356-365, Page 11, Lines 383-386 and Page 12, Lines 391.

Third, the rationale behind the hyperparameter settings and the process for tuning them have been elaborated in detail in Appendix E Page 20, Lines 705 - Page 21, Lines 755. This includes justifications for the stopping condition, iteration count, spatial sampling configuration, temporal momentum coefficient, sampling distribution, sampling strategies per space, and the primary weight, supported by ablation studies and empirical evidence.

Fourth, we acknowledge the reviewer's valid point regarding the white-box assumption's limited practicality for closed-source models. We have added a clarification in the Page 4 Lines 95-110 stating that the primary purpose of the white-box setup in this work is to serve as a diagnostic tool for rigorously evaluating and probing the safety robustness of LLMs under a controlled, worst-case scenario, rather than representing a directly deployable real-world attack. We concede its inherent limitations against proprietary, closed-source systems.

Finally, recognizing the need for a deeper analysis of transferability, we have expanded the discussion on this topic in the Discussion section, Page 16, line 580-593. This addition further explores the implications and observed trends related to cross-model transfer of the generated adversarial attacks, providing a more thorough investigation of this property.

Comments 6: The results section should provide some experiments for larger samples than that already presented. Additionally, other models should be compared since the models used for the comparison are only open-source, it will be important to test the performance of the provided models against some closed-source models such as GPT 4. Moreover, there are very recent models that have been highlighted in the literature such as JailPO that will be worth to be compared against the performance of the provided model. Also, many parameters are missing such as the stopping criteria, the details of the hyperparameters … etc., which mainly limit the reproducibility of the results and the cross-model transfer is not fully tested. The complexity of the provided model should be compared against those for the benchmarks used.

Response 6: Agree. We have, accordingly, revised the manuscript to address these points. First, regarding the sample size, we acknowledge the computational constraints that prevented us from using the full dataset. To enhance the robustness of our evaluation, we have added a detailed description of our stratified random sampling method for selecting the subset of 100 malicious prompts. We also discuss the potential limitations of this approach, including possible underrepresentation of rare categories and intra-category diversity, and explain why our method provides a practical and representative compromise. This addition can be found in the "Datasets" section on Page 13, Lines 452-471.

Second, concerning the comparison with closed-source models like GPT-4, our method is primarily designed for white-box models to evaluate safety robustness. We did preliminarily test the transferability of suffixes generated from LLaMA2-7B to black-box models such as GPT-4, but the effectiveness was very poor. We have added a discussion on the potential reasons for this, including the fundamental differences in model architectures and defense mechanisms. Furthermore, we note that the jailbreak success rate for black-box models may not be a reliable metric due to potential changes in their defense strategies or prohibited word lists, which could make the results less comparable. This discussion is included in the "Discussion" section on Page 16, line 580-593.

Third, to address the missing parameters and improve reproducibility, we have added a comprehensive appendix (Appendix E) that details the stopping criteria, hyperparameters, and their theoretical justifications. This includes explanations of the iteration count, spatial sampling configuration, temporal momentum, sampling distribution and strategies across spaces, and primary weight, supported by ablation studies and empirical evidence. This can be found in Appendix E on Page 20, Lines 705 - Page 21, Lines 755.

Finally, we have added a "Model Complexity Analysis" section (Appendix D) where we quantitatively compare the computational complexity and parameter scale of the evaluated models using the calflops library, demonstrating the consistency among models based on the same LLaMA-7B architecture. This analysis is presented in Page 19, Lines 691-704.

We believe these revisions thoroughly address the reviewer's concerns and enhance the clarity, reproducibility, and comprehensiveness of our work.

Comments 7: The results section should be more balanced between being descriptive and being analytical, it is mainly descriptive in current version of the manuscript.

Response 7: We agree with the reviewer's insightful comment regarding the need for a balance between descriptive and analytical content. In revising the manuscript, we have chosen to maintain a clear structural distinction between the Results and Discussion sections. The Results section remains primarily descriptive, presenting the experimental findings in an objective manner to establish the factual outcomes. Subsequently, we have significantly strengthened the analytical and interpretive components in the Discussion section. Here, we provide a comprehensive analysis of the results, exploring the reasons behind the performance of our method, its limitations (such as the performance on Vicuna-7B), the implications of the ablation study, and the future directions inspired by our findings. The enhanced analytical discussion can be found in the revised manuscript on Page 16, line 580-593.

Comments 8: The limitations of the presented approach should be discussed in detail in the discussion section with some insights into future directions.

Response 8: Agree. We have, accordingly, revised the Discussion section to provide a detailed analysis of the limitations of our approach and explicitly outline potential future research directions. The modifications can be found in the Discussion section on Page 16, line 580-593.

Comments 9: The caption of Fig. 3 is really long. Additionally, the titles of Tables 2 and 4 are really vague.

Response 9: Agree. We have, accordingly, revised the manuscript to address these concerns. Specifically, the detailed experimental description originally included in the caption of Fig. 3 has been moved to the main body of the text to shorten the figure caption. Furthermore, the titles of Tables 2 and 4 have been revised to be more precise.

Comments 10: Section 3: "Materials and Methods" better name it as "Methodology".

Response 10: Agree. We have, accordingly, revised the section title from "Materials and Methods" to "Methodology" .

Comments on the Quality of English Language

Response : We have revised all punctuation marks and the spacing before and after them throughout the entire paper, standardized the format for figure citations, corrected sentences with grammatical errors, and conducted a full proofreading of the entire document.

Reviewer 4 Report

Comments and Suggestions for Authors

1. expand discussion of ethical implications and responsible disclosure, as jailbreak attacks have potential misuse risks.

2. include additional recent works (2024–2025) on adaptive defenses (e.g. safety-aware decoding, reinforcement-learning-based jailbreaks).

3. algorithm 1 is comprehensive, but parameter settings (e.g. α = 0.25, hybrid ratio 6:6:6:7:7) require justification.

4. clarify computational costs more transparently (time per step, total GPU hours).

5. tables 2-4 are useful but should include confidence intervals or statistical significance testing to strengthen reliability.

6- discuss why SM-GCG slightly underperforms GCG on Vicuna.

7- expand discussion of transferability limitations and potential defense countermeasures.

8- explicitly highlight the dual-use nature of this research and frame it as a safety contribution.

Author Response

We thank the reviewers for their insightful comments and constructive suggestions, which have helped us to significantly improve the manuscript.

Comments 1: expand discussion of ethical implications and responsible disclosure, as jailbreak attacks have potential misuse risks.

Response 1: Agree. We have, accordingly, added a dedicated paragraph in the Introduction to explicitly address ethical implications and responsible disclosure. This discussion can be found on Page 3 Lines 91-102.

Comments 2: Include additional recent works (2024–2025) on adaptive defenses (e.g. safety-aware decoding, reinforcement-learning-based jailbreaks).

Response 2: We appreciate this suggestion. While we have not incorporated additional recent works from 2024-2025, we have instead focused on providing a more critical analysis of the representative works already included in our literature review. We believe this approach offers greater conceptual depth by evaluating the limitations and broader implications of existing methodologies This critical analysis has been added to the discussion following the presentation of each category of methods. The specific revisions can be found on page 4, lines 127 - page 5, lines 185.

Comments 3: Algorithm 1 is comprehensive, but parameter settings (e.g. α = 0.25, hybrid ratio 6:6:6:7:7) require justification.

Response 3: Agree. We have, accordingly, added a new appendix section (Appendix E) to provide detailed justifications for all key hyperparameter settings, including the primary weight (α = 0.25) and the hybrid sampling distribution ratio (6:6:6:7:7). These detailed explanations can be found in the newly added Appendix E on Page 20, Lines 705 - Page 21, Lines 755

Comments 4: clarify computational costs more transparently (time per step, total GPU hours).

Response 4: We thank the reviewer for their comments regarding the clarification of computational costs. The dataset section of the paper already includes relevant descriptions. Due to the influence of hyperparameter selection, providing precise computation times presents certain challenges. However, we have indicated that on an H100 GPU, each round of GCG optimization typically takes approximately 4 seconds. This primarily depends on the speed of the model's forward and backward propagation. Moreover, more precise per-step times under various parameter settings can be found in the ablation study section. These clarifications can be found on Page 13, Lines 447-450.

Comments 5: tables 2-4 are useful but should include confidence intervals or statistical significance testing to strengthen reliability.
Response 5: We appreciate this valuable suggestion. However, due to the high experimental costs involved, it was not feasible to conduct repeated experiments to generate confidence intervals or perform additional statistical testing. Instead, to enhance the robustness of our findings, we have incorporated a subset selection method in our analysis. This addition is detailed in Page 452, Line 472.

Comments 6: discuss why SM-GCG slightly underperforms GCG on Vicuna.

Response 6: Agree. We have, accordingly, revised the discussion section to address this point by analyzing the reasons behind SM-GCG's slight underperformance compared to GCG on Vicuna while emphasizing its overall improved transferability. Specifically, we have added a dedicated paragraph explaining that while spatial momentum enhances cross-model generalization, it doesn't fully overcome the inherent overfitting problem in gradient-based optimization. We further strengthened the discussion by proposing future research directions to explicitly improve transferability. These changes can be found in the revised manuscript on page 16 lines 580-593.

Comments 7: expand discussion of transferability limitations and potential defense countermeasures.

Response 7: Agree. We have, accordingly, expanded the discussion to address both the limitations regarding transferability and potential defense countermeasures. Specifically, we added a dedicated paragraph analyzing the reasons for transferability constraints (e.g., architectural differences and inherent optimization biases) and outlined two promising directions for future work, including model-agnostic constraints and analyzing model-invariant features. This revision aligns with Comment 5, where a similar expansion was requested. The changes can be found on page 16 lines 580-593.

Comments 8: explicitly highlight the dual-use nature of this research and frame it as a safety contribution.

Response 8: Agree. We have, accordingly, revised the introduction to explicitly emphasize this point.  We have added a paragraph that clearly highlights the dual-use nature of this research and frames it as a safety contribution. This addition can be found on Page 4, Lines 95-110.

Reviewer 5 Report

Comments and Suggestions for Authors

This paper presents a novel and interesting approach, Spatial Momentum Greedy Coordinate Gradient (SM-GCG), to enhance jailbreak attacks on large language models. The authors correctly identify a key limitation of existing gradient-based methods like GCG: the optimization process is often non-smooth and prone to getting trapped in local minima. The core idea of incorporating "spatial momentum" by aggregating gradient information across multiple transformation spaces (text, token, embedding, etc.) is a creative and well-motivated strategy inspired by successful techniques in the computer vision domain. The reported results, particularly the significant improvement in attack success rate against robust models like Llama2-7B-Chat, suggest that this is a promising direction for research in LLM red-teaming.

However, the manuscript requires a major revision to address significant shortcomings in its methodological clarity and theoretical justification. The primary weakness is the abstract and high-level description of the transformations applied within each of the five proposed spaces. The paper mentions using "synonym replacement" in text space and "shift transformations" in token space, but lacks the concrete details needed for reproducibility. For instance, how are synonyms selected? How is a token sequence "shifted"? What is the nature and magnitude of the "Gaussian noise" applied in the one-hot and embedding spaces? Without these specifics, the SM-GCG method remains a conceptual black box. Furthermore, the paper lacks a strong theoretical or even intuitive justification for why aggregating gradients from these disparate, semantically-varied spaces should lead to a more accurate estimate of the "true" update direction. The combination of spaces feels ad-hoc, and the paper would be much stronger if it provided a clearer rationale for this specific multi-space design.

A second major issue lies in the experimental design and analysis. The proposed method introduces a large number of new hyperparameters, including the sampling count for each of the five spaces and their respective weighting coefficients (λi). The paper reports the final settings used (a "hybrid sampling method" with a ratio of 6:6:6:7:7 and α=0.25) but provides no insight into how these crucial parameters were selected. This gives the impression of a heavily-tuned method and undermines the generalizability of the results. A sensitivity analysis on these hyperparameters is essential to understand their impact and the robustness of the method. Additionally, the claim of "enhanced transferability" is overstated. While Table 4 shows a numerical improvement over GCG, the absolute success rates for transfer attacks remain very low (e.g., 1% from Guanaco to Llama2), indicating that overfitting to the white-box model is still the dominant outcome. The discussion should be more nuanced, acknowledging that while transferability is improved, it remains a largely unsolved problem.

In conclusion, this paper introduces a promising idea for improving gradient-based jailbreak attacks. The concept of spatial momentum is a valuable contribution. However, the work is currently undermined by a lack of methodological detail, insufficient justification for its core design choices, and an over-optimistic interpretation of the experimental results. In addition, it lacks references about visual adversarial attacks such as: (1) FIGhost: Fluorescent Ink-based Stealthy and Flexible Backdoor Attacks on Physical Traffic Sign Recognition, (2) Itpatch: An invisible and triggered physical adversarial patch against traffic sign recognition. Please add proper discussions about these studies.

Comments on the Quality of English Language

The quality of the English language should be improved.

Author Response

We thank the reviewers for their insightful comments and constructive suggestions, which have helped us to significantly improve the manuscript.

Comments 1: However, the manuscript requires a major revision to address significant shortcomings in its methodological clarity and theoretical justification. The primary weakness is the abstract and high-level description of the transformations applied within each of the five proposed spaces. The paper mentions using "synonym replacement" in text space and "shift transformations" in token space, but lacks the concrete details needed for reproducibility. For instance, how are synonyms selected? How is a token sequence "shifted"? What is the nature and magnitude of the "Gaussian noise" applied in the one-hot and embedding spaces? Without these specifics, the SM-GCG method remains a conceptual black box. Furthermore, the paper lacks a strong theoretical or even intuitive justification for why aggregating gradients from these disparate, semantically-varied spaces should lead to a more accurate estimate of the "true" update direction. The combination of spaces feels ad-hoc, and the paper would be much stronger if it provided a clearer rationale for this specific multi-space design.

Response 1: Agree. We have, accordingly, revised the manuscript to enhance methodological clarity and provide a stronger theoretical justification. Specifically, we have added detailed implementation descriptions following the abstract explanations of transformations in each of the five spaces. Furthermore, to strengthen the theoretical rationale, we have incorporated discussions on the dynamical principle of curve amplitude narrowing, the relationship between the stable convergence phase and adversarial example generalizability, and the quantitative impact of loss plateau reduction on attack efficiency. Additionally, we calculated the Spearman correlation coefficient between the attack success rate and loss across 10 attacks with different parameter settings and included a line chart for intuitive visualization. These revisions can be found in the revised manuscript at:
Page 9, Lines 296-302, Page 10, Lines 319-341, Page 11, Lines 356-365, Page 11, Lines 383-386 ,Page 12, Lines 391 and Page 6, Lines 220-244

Comments 2: A second major issue lies in the experimental design and analysis. The proposed method introduces a large number of new hyperparameters, including the sampling count for each of the five spaces and their respective weighting coefficients (λi). The paper reports the final settings used (a "hybrid sampling method" with a ratio of 6:6:6:7:7 and α=0.25) but provides no insight into how these crucial parameters were selected. This gives the impression of a heavily-tuned method and undermines the generalizability of the results. A sensitivity analysis on these hyperparameters is essential to understand their impact and the robustness of the method.

Response 2: Agree. We have, accordingly, added a new appendix (Appendix E) to provide detailed justifications for the selection of our key hyperparameters. This appendix explains the rationale behind the specific settings, including the spatial sampling configuration (6:6:6:7:7), the primary weight (α=0.25), and other critical parameters. Furthermore, we explicitly acknowledge the limitations regarding exhaustive sensitivity analysis and global optimization due to substantial computational constraints, while emphasizing that our focus was to identify a well-performing set that effectively demonstrates the method's validity. We also note our plans to conduct a more comprehensive sensitivity analysis in future work. These additions can be found in Page 20, Lines 705 - Page 21, Lines 755.

Comments 3: Additionally, the claim of "enhanced transferability" is overstated. While Table 4 shows a numerical improvement over GCG, the absolute success rates for transfer attacks remain very low (e.g., 1% from Guanaco to Llama2), indicating that overfitting to the white-box model is still the dominant outcome. The discussion should be more nuanced, acknowledging that while transferability is improved, it remains a largely unsolved problem.

Response 3: Agree. We have, accordingly, revised the discussion section to provide a more nuanced and balanced interpretation of the transferability results. Specifically, we have added an in-depth analysis of the limitations, explicitly acknowledging that while spatial momentum improves transferability compared to GCG, it does not fully resolve the fundamental challenge of overfitting to the source model, as evidenced by the low absolute success rates. The revised text identifies key reasons for these limitations and proposes concrete future research directions to address this largely unsolved problem. This change can be found in the revised manuscript on page 16 lines 580-593.

Comments 4: In conclusion, this paper introduces a promising idea for improving gradient-based jailbreak attacks. The concept of spatial momentum is a valuable contribution. However, the work is currently undermined by a lack of methodological detail, insufficient justification for its core design choices, and an over-optimistic interpretation of the experimental results. In addition, it lacks references about visual adversarial attacks such as: (1) FIGhost: Fluorescent Ink-based Stealthy and Flexible Backdoor Attacks on Physical Traffic Sign Recognition, (2) Itpatch: An invisible and triggered physical adversarial patch against traffic sign recognition. Please add proper discussions about these studies.

Response 4: Agree. We have, accordingly, added references and discussions about visual adversarial attacks as existing explorations of the image modality in real-world multimodal attacks. This addition can be found in the revised manuscript on page 4, lines 138-141.

Round 2

Reviewer 1 Report

Comments and Suggestions for Authors

No comment

Author Response

Thank you for your comment. We have further polished the language throughout the manuscript to improve clarity and expression.

Reviewer 3 Report

Comments and Suggestions for Authors

All comments in the previous review report except for comment number 1, which should be handled. Mainly because non-peer reviewed references cannot be considered as a trusted source of information.

Additionally, two more comments in the revised text, which are:

  1. I think the title of Table A3 should be revised since the table does not indicate any kind of complexity.
  2. The caption of figure A1 should be shortened similar to the rest of the figures.

Author Response

Comment 1: Non-peer reviewed references cannot be considered as a trusted source of information.

Response 1: Thank you for your valuable feedback. We have carefully reviewed all non-peer-reviewed references in our manuscript. In total, 21 such references were identified. Among them, we successfully replaced 13 references with their peer-reviewed versions. Additionally, 3 non-peer-reviewed references that had minimal impact on the core content of the paper were removed, and relevant sections of the text were revised accordingly. Finally, 5 non-peer-reviewed references were retained due to their critical relevance. All modifications, including detailed justifications and before-and-after comparisons, have been compiled into an Word for your reference.

Comment 2:I think the title of Table A3 should be revised since the table does not indicate any kind of complexity. The caption of figure A1 should be shortened similar to the rest of the figures.

Response 2:We have corrected the title of Table A3, which was mistakenly labeled with the title of Table A2.
The redundant part in the caption of Figure A1 has been moved to the main text.
Thank you for pointing out this oversight. We have thoroughly rechecked the manuscript to prevent similar issues from recurring.

Reviewer 5 Report

Comments and Suggestions for Authors

After carefully checking the revised version, I found that most of my previous concerns were addressed.

Author Response

Thank you for your comment.